# Effects of *Cordyceps militaris* Extracts on Macrophage as Immune Conductors

**Yoonjin Park** [1,2,†], **Seoyoon Choi** [1,†], **Boyong Kim** [1,2,3,*] and **Seung Gwan Lee** [1,*]

1   Department of Clinical Laboratory Sciences, College of Health Science, Korea University, Seongbuk-gu, Seoul 02841, Korea; pyoonjin@naver.com (Y.P.); zenna_yoon@naver.com (S.C.)
2   Life Together, 13 Gongdan-ro, Chuncheon-si, Gangwon 24232, Korea
3   Mitosbio, 13 Gongdan-ro, Chuncheon-si, Gangwon 24232, Korea
*   Correspondence: erythro74@korea.ac.kr (B.K.); seunggwan@korea.ac.kr (S.G.L.)
†   These authors contributed equally to this work.

**Abstract:** Although *Cordyceps militaris* is documented several medicinal functions, there is not enough for demonstration of leukocytic differentiation. Cordycepin and adenosine were 11.75 µg and 1.25 µg in the extract, respectively. Unlike the levels of TNF-α and IL-1β in macrophages that were approximately 4 time and 48 times higher than the control under lipopolysaccharides (LPS), macrophages under the extract (1 µg/mL) showed 13- and 10-fold lower TNF-α and IL-1β levels than the LPS-treated cells. This was corroborated by flow cytometry, where their levels were 20 times and 14 times lower, respectively. Under the extract, the LPS-treated macrophages enhanced M2 polarization and attenuated M1 polarization. In addition, the extract also dose-dependently activated macrophage phagocytosis. Under the extract conditioned medium, dendritic cells (DCs) were strongly differentiated toward CD11b[+] and Xcr1[+] cells because their densities were 13.6 times and 6.26 times higher than those in the LPS conditioned medium, respectively. Differentiation of $T_{reg}$ and natural killer T-like (NKTL) cells also were increased about 1.67 times and 6.73 times than those in the LPS conditioned medium, respectively. These results suggest that the *C. militaris* extract has strong effects on the modulation of macrophages and dendritic cells and T cells under inflammatory stress.

**Keywords:** *Cordyceps militaris*; cordycepin; inflammation; dendritic cell; macrophage; T cell

## 1. Introduction

*Cordyceps* species have diverse biological activities and are well known in Chinese traditional medicine. The *Cordyceps* genus contains around 400 species worldwide. Among them, *Cordyceps militaris* and *Cordyceps sinensis* have been traditionally used since ancient times, and are distributed worldwide [1,2].

Cordycepin, or 3′-deoxyadenosine, a functional compound found in some *Cordyceps*, is a precursor of polysaccharides, ergosterol, mannitol, and vitamins [3]. Cordycepin is produced by *C. militaris*, *C. kyusyuensis*, and *Aspergillus nidulans* [4]; it was first discovered in 1951 by Professor Cunningham of the University of Glasgow, United Kingdom, in *C. militaris* [5] and *Aspergillus nidulans* phylogenetically distant *C. militaris* [6]. The pharmacological functions of cordycepin are very diverse in vivo and in vitro; it shows anti-cancer, anti-inflammatory, anti-viral, anti-leukemia, anti-tumor, anti-diabetic, and anti-obesity effects and is known to modulate the human immune system [7,8]. Cordycepin has various biological effects, including inhibition of mRNA polyadenylation and mRNA elongation, activation of AMP-activated protein kinase (AMPK), and protein phosphatase [9–11]. Additionally, *Cordyceps* was documented as the therapeutic agent associated with anti-inflammation, immunoregulation, anti-hypoglycemia, renal protection, anti-tumor, anti-diabetes, antioxidant, anti-virus, and cardiovascular protection [12,13]. However, there are side effects of *Cordyceps*, including increasing autoimmune diseases,

slow blood clotting, and interaction with other drugs [14]. Although there are no clear causes for the side effects, *Cordyceps* has many bioactive compounds, including polysaccharides, terpenes, phenolic compounds, and proteins [13]. The suggested dose of *Cordyceps* is two and three times with 1050 mg extract or 0.14% adenosine [14].

Macrophages, which are derived from monocytes, play an important role in phagocytosis in disease lesions and act as anti-inflammatory and pro-inflammatory cells depending on their polarization [15]. They are also involved in the modulation of innate and acquired immunities [15]. During inflammatory reactions, they also play a defensive role by producing nitric oxide (NO) and cytokines [16]. Lipopolysaccharides (LPS) are major endotoxins that are derived from gram-negative bacterial cell walls [17]. Macrophages exposed to LPS produce inflammatory cytokines, including interleukin (IL)-1β, IL-6, and tumor necrosis factor (TNF), and they polarize to M1 macrophages [18,19]. Although reported researches have shown the anti-inflammatory effects of *C. militaris* extracts [20], there is no clear evidence of whether these extracts could act on macrophages to conducting the immune response.

The goal of this study was to quantify the 3′-deoxyadenosine levels in *C. militaris* and to evaluate the biological functions of *C. militaris* extracts in the immunological conducting of LPS-treated macrophages.

## 2. Materials and Methods

### 2.1. Measurement of Cordycepin by High-Performance Liquid Chromatography (HPLC)

Cordycepin (3′-deoxyadenosine, ≥98.0%, HPLC grade, product number C3394-10MG, CAS Number 73-03-0) and adenosine (≥99%, HPLC grade, P no. A9251-1G, CAS Number 58-61-7) were purchased from Sigma Aldrich (St. Louis, MO, USA). They were used for HPLC (Dionex Ultimate 3000, Thermo Fisher Scientific Inc., Germering, Germany) and the retention times for cordycepin and adenosine were 6.923 min and 5.687 min, while the linear equations for their standard curves were $y = 0.7745x - 0.0528$ ($r^2 = 0.9993$) and $y = 0.5606x - 0.1491$ ($r^2 = 0.9993$), respectively. The standard solutions of cordycepin and adenosine were injected into the HPLC instrument to evaluate the limits of detection (LOD) and quantification (LOQ), calculated by the formulae $LOD = 3\sigma/S$ and $LOQ = 10\sigma/S$ (σ: standard deviation of the response, S: slope of the calibration plot).

### 2.2. Extraction of Cordycepin from Cordyceps militaris

Two grams of *C. militaris* powder (LIFE TOGETHER CO., LTD and MITOS BIO, INC., Chuncheon, Korea) was mixed with distilled water and treated at 85 °C for 2.5 h, followed by ultrasonic extraction at 600 W for 35 min. The extract was filtered through a microporous membrane (0.22 μm, Merck, Darmstadt, Germany) and analyzed using HPLC to quantify cordycepin levels. Chromatographic separation was performed using the YMC-Triart C18 column (TA12S05-2546WT, product number 97049-920) under the following conditions: mobile phase: methanol (15%)–water (85%); analytical time: 30 min; column temperature: 25 °C; flow rate: 1.0 mL/min; UV detection wavelength: 260 nm; injection volume: 10 μL. The entire cycle was repeated thrice.

### 2.3. Cell Culture

RAW 264.7 cells, a murine macrophage cell line, was maintained in a 5% $CO_2$ atmosphere in Dulbecco's modified Eagle's medium (DMEM) supplemented with 10% heat-inactivated fetal bovine serum (FBS), 100 μg/mL penicillin, and 100 μg/mL streptomycin. Cells were exposed to four conditions including LPS (100 ng/mL), the extracts (Ext), cordycepin (Cor, 30 μM) (Sigma), pre-LPS, and post-extract (LPS+Ext). Cells were exposed to LPS for 12 h and cleaned with PBS (pH 7.4). Subsequently, the cleaned cells were treated with *C. militaris* extracts (0.1, 0.25, 0.5, or 1 μg/mL) for 8 h. After exposing, the supernatants from the four conditions were collected. To analyze cellular differentiation, monocytes (JAWS Ⅱ; ATCC CRL-11904) and T cells (TK-1; ATCC CRL-2396) were cultured with the four conditioned media derived from the supernatant of macrophages treated with

Ext1 (extract 1 µg/mL), LPS, LPS+Ext1, or 0.5 and Cor 30 µM. Additionally, to estimate the differentiating capacity of the extract, the results of reference were evaluated using Dynabeads, CD3/CD28 (Thermo Fisher Scientific Inc.).

For precise evaluation, the results for samples under the conditioned media were compensated with results for treating with the same as the concentration of the LPS, extracts, and cordycepin (Sigma) in the treated supernatant.

### 2.4. Real-Time Polymerase Chain Reaction (PCR)

Total RNA was extracted from cells using Ribospin mini kit (GeneAll, Seoul, Korea); then, RNA samples were converted to cDNA using DiaStarTM 2X RT Pre-mix kit (Solgent, Daejeon, Korea) at 55 °C for 1 h and then at 95 °C for 5 min. Real time-PCR was performed using TOPreal ™ qPCR 2X PreMIX (Enzynomics; Daejeon, Korea) with the cDNA samples and analyzed using the StepOnePlus Real-Time PCR System instrument (Thermo Fisher Scientific Inc., Germering, Germany). TNF-α, IL-1β, and b-actin primers (Table 1) were purchased from Bioneer (Seoul, Korea). PCR conditions were as follows: initial denaturation of 15 min at 95 °C, followed by 35 cycles of 10 s at 95 °C, 15 s at 60 °C, and elongation of 30 s at 72 °C. Data were analyzed using the StepOne Software V2.3.

**Table 1.** Primers used for real-time PCR.

| Gene | | Sequence |
|---|---|---|
| *β-actin* | Forward | 5′-GTG GGV CGC CCT AGG ACC AG-3′ |
| | Reverse | 5′-GGA GGA AGA GGA TGC GGC AGT-3′ |
| *TNF-α* | Forward | 5′-TTG ACC TCA GCG CTG AGT TA-3′ |
| | Reverse | 5′-CCT GTA GCC CAC GTC GTA GC-3′ |
| *IL-1β* | Forward | 5′-CAG GAT GAG GAC ATG ACA CC-3′ |
| | Reverse | 5′-CTC TGC AGA CTC AAA CTC CAC-3′ |

### 2.5. Flow Cytometric Analysis

Cells treated with LPS and *C. militaris* extracts (0.1, 0.25, 0.5, and 1 µg/mL) were fixed with 2% paraformaldehyde for 2 h at 40 °C [21]. Fixed cells were treated with Tween-20 (0.5%) and incubated with the specific probes (TNF-α; FAM-5′-UUGACCUCAGCGCUGAG-UUA-3′ and IL-1β; TAMRA-5′-CAGGAUGAGGACAUGACACC-3′). Then, cells were analyzed using BD FACSCalibur (BD science, San Jose, CA, USA) [22]. To analyze macrophage polarization, macrophages treated with 0.02% Tween-20 were incubated with the following probes: CXCL10 (Alexa fluor 647-5′-GCTTCCAAGGATGGACCACA-3′) and CCL22 (Alexa fluor 670-5′-GAGATCTGTGCCGATCCCAG-3′). Stained cells were analyzed with a flow cytometer BD FACSCalibur and FlowJo 10.6.1 (BD Bioscience, Gangnam-gu, Seoul). To analyze cellular differentiation, monocytes and T cells were treated with fluorescence-conjugated antibodies including FITC-anti-CD11b (M1/70, monoclonal, rat) (BioLegend, San Diego, CA, USA), PE- anti-Xcr1 (ZET, monoclonal, mouse) (BioLegend), APC- anti-CD304 (3E12, monoclonal, mouse) (BioLegend), FITC- anti-FoxP3 (FJK-16s, monoclonal, mouse) (Thermo Fisher Scientific Inc.), and PE- anti-NK1.1 (PK136, monoclonal, mouse) (BioLegend).

### 2.6. Phagocytic Activity

After treatment with LPS for 12 h and *C. militaris* extracts, macrophages were exposed to fluorescence-labeled *Escherichia coli* particles for 8 h and phagocytosis was examined using Vybrant™ Phagocytosis Assay Kit (Thermo Fisher Scientific Inc., Germering, Germany) for 2 h. Stained macrophages were analyzed by flow cytometry (BDcalibur and FlowJo 10.6.1, BD Bioscience).

### 2.7. Statistical Analysis

All experiments were three independent experiments with estimating five times per one experiment (n = 3), and the data were analyzed by one-way analysis of variance (ANOVA) with post hoc test (Scheffe's method), and a sample without normality was analyzed using a non-parametric test (Kruskal–Wallis H test) with the SPSS v26 (IBM, New York, NY, USA) and Prism 7 (GraphPad, San Diego, CA, USA) software.

## 3. Results

This research is to document the biological functions of *Cordyceps militaris* extract between macrophages and leukocytes such as dendritic cells and T cells. Under the extract, macrophages were activated their polarization and differentiation of dendritic cells and T cells also were activated under the conditioned medium (Figure 1).

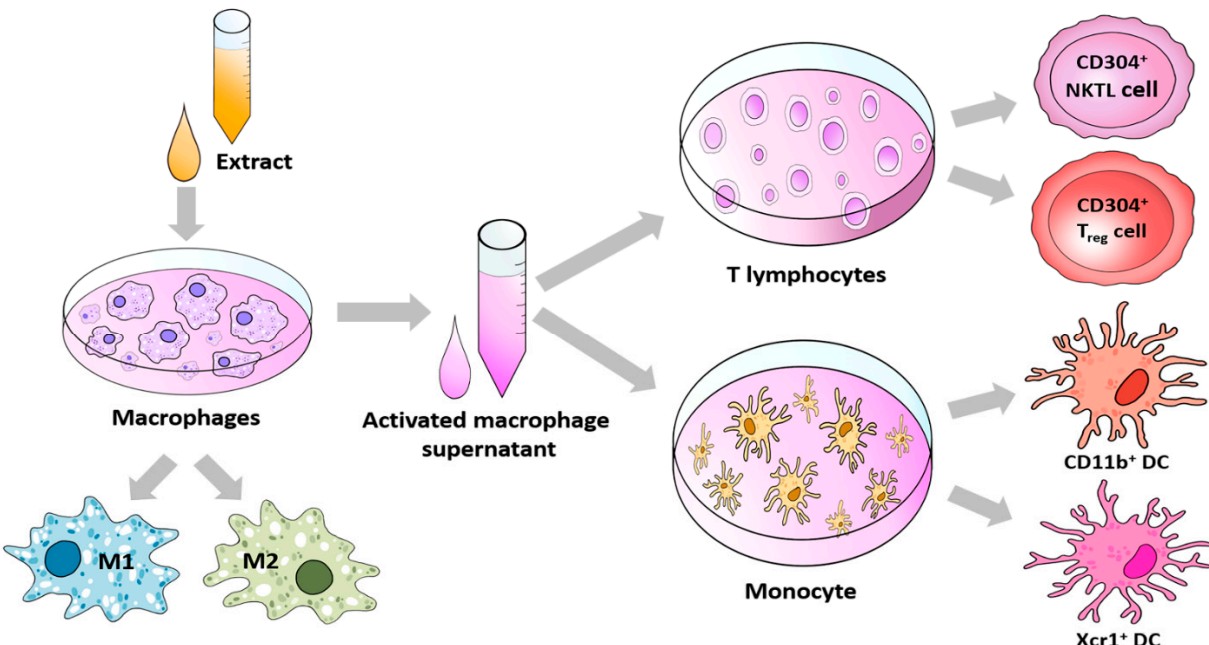

**Figure 1.** Schematic workflow for experimental procedures. The macrophages were exposed to the extract for the polarization. Under the extract, the supernatants derived from macrophages were collected and exposed to T cells and monocytes. M1 and 2; macrophage 1 and 2, DC; dendritic cells, NKTL cells; natural killer T-like cells.

### 3.1. Measurement of Cordycepin Levels in Cordyceps militaris Extracts

Cordycepin and adenosine levels were quantified by HPLC using appropriate standards (Figure 2a). The total cordycepin and adenosine concentrations in the extracts were $0.47 \pm 0.09$ μg/mL and $0.05 \pm 0.01$ μg/mL, respectively. Reported to the mycelium weight, the concentrations of cordycepin and adenosine in the fresh mycelium were 11.75 μg/g and 1.25 μg/g, respectively (Figure 2a,b).

### 3.2. Effects of Cordyceps militaris Extracts on the Expression of IL-1β and TNF-α in Macrophages

The mRNA levels of pro-inflammatory cytokines TNF-α and IL-1β were quantified in RAW 264.7 cells treated with LPS and extracts containing cordycepin. In macrophages stimulated only with LPS, the mRNA levels of TNF-α and IL-1β increased approximately four- to five-folds compared with the control (Figure 3a). For macrophages treated with LPS and cordycepin extracts, the mRNA levels of TNF-α and IL-1β dramatically decreased, especially for 1 μg/mL of extract (Figure 3a). Fluorescence in situ hybridization results also reveal that TNF-α and IL-1β were downregulated dose-dependently in macrophages

treated with these extracts (Figure 3b). Notably, treatment with 1 µg/mL cordycepin extract dramatically downregulated these cytokine levels (Figure 3).

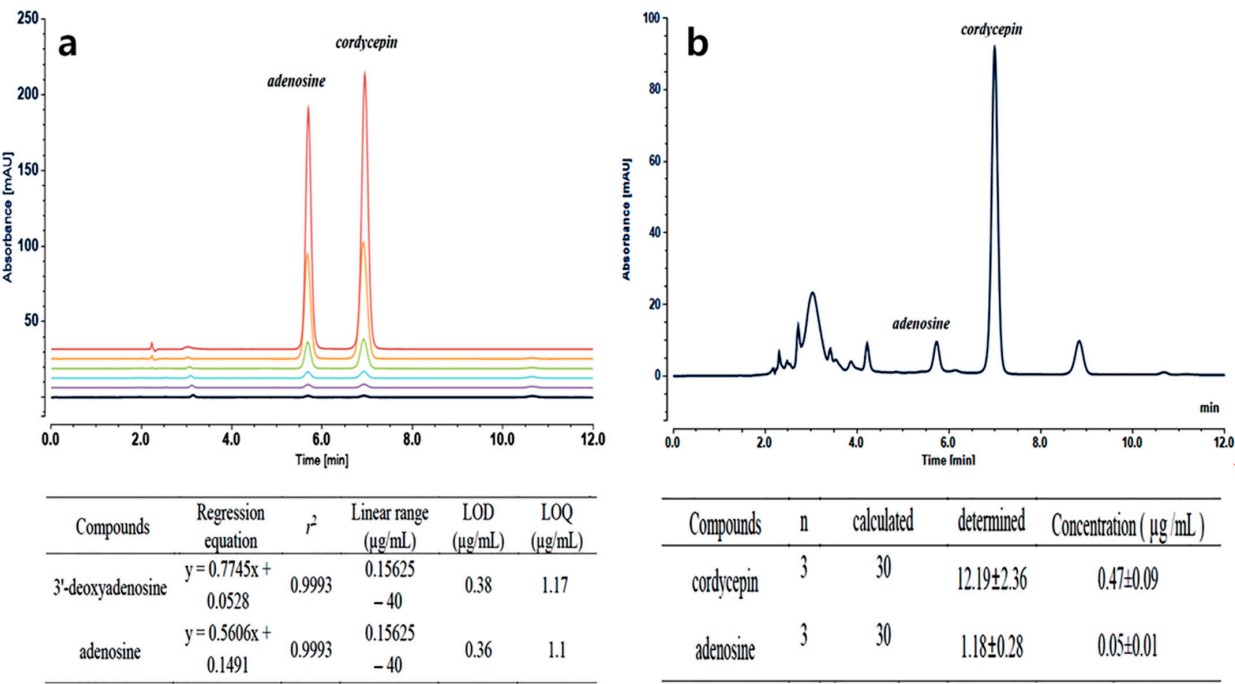

**Figure 2.** Determination of the cordycepin content in *Cordyceps militaris* extracts by high-performance liquid chromatography (HPLC). Panel (**a**) showed the overlay chromatogram of various concentrations of the cordycepin and adenosine standards. Panel (**b**) showed the concentration of cordycepin in *Cordyceps militaris* extracts. ($p < 0.05$).

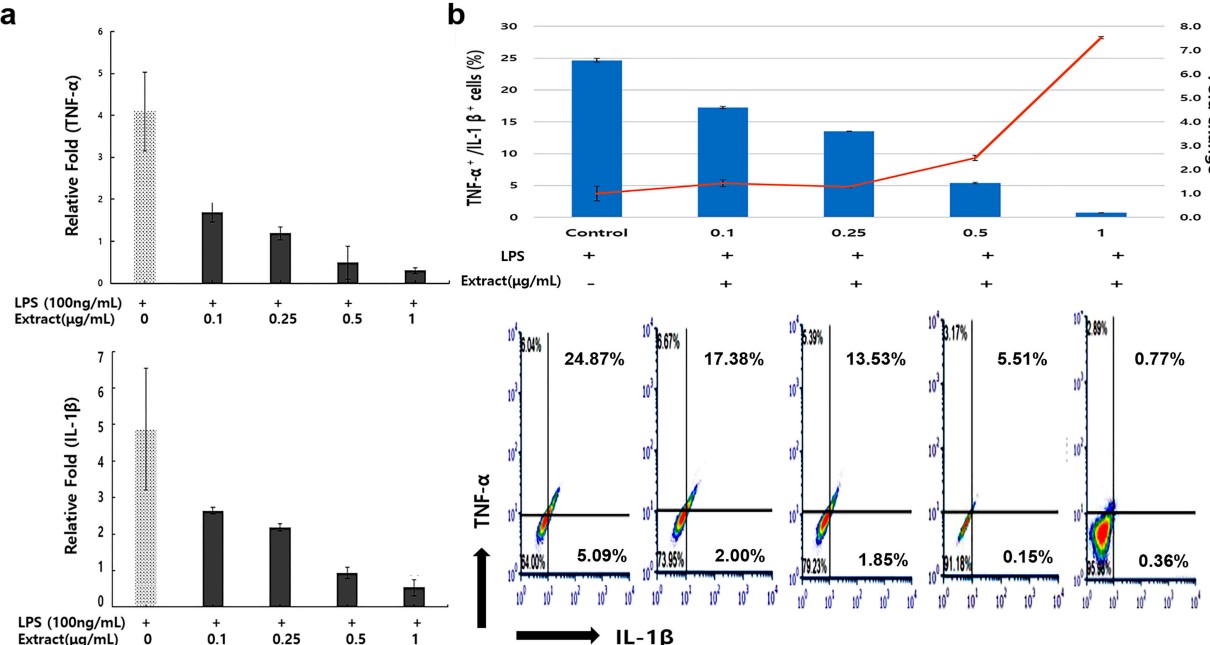

**Figure 3.** Expression levels of IL-1β and TNF-α from macrophages activated by *Cordyceps militaris* extracts under lipopolysaccharides (LPS) stress. Expression levels of IL-1 β and TNF-α from macrophages using qPCR and FACSCalibur have been shown in the (**a**,**b**) panels, respectively. The bar graphs shown in the (**b**) panel are the relative fold changes of double-positive cells by the doses of the extract under LPS stress. ($p < 0.05$).

### 3.3. Polarization of Macrophages

For cells not treated with the cordycepin extract, the number of M1 macrophages increased approximately four times following stimulation with LPS; however, under LPS treatment, the number of M2 macrophages decreased by approximately 60% compared to the control (Figure 4). Surprisingly, when treated with *C. militaris* extracts, the number of M1 macrophages decreased and that of M2 macrophages increased dose-dependently (Figure 4). When 1 µg/mL extract was used, the number of M1 macrophages decreased approximately four times, and the number of M2 macrophages increased approximately nine times compared to the number observed following LPS treatment (Figure 4).

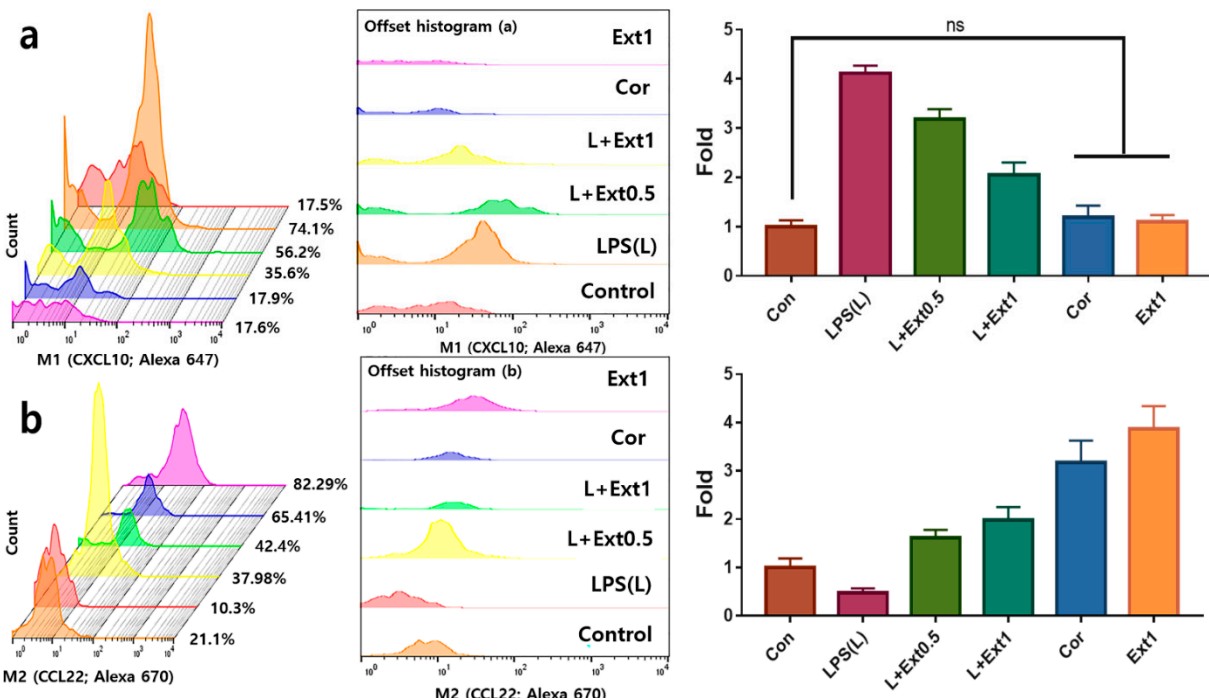

**Figure 4.** Polarization patterns of macrophages by *Cordyceps militaris* extracts under LPS stress. Fluorescence in situ hybridization (FISH) results measured by flow cytometry. For the histograms of M1 and M2 labeled cells, populations of macrophages were gated. Changes in M1 and M2 polarization have been shown in panel (**a**,**b**), respectively. Bar graphs show the relative fold changes for M1 and M2 polarization by the doses of the extract under LPS stress. ns; not significant, LPS; lipopolysaccharide, Ext; extract, Cor; cordycepin. ns; not significant, (all data are $p < 0.05$ without ns).

### 3.4. Phagocytic Activity of Macrophages

Changes in the phagocytic activity were not clear when macrophages were treated only with LPS; however, when cells were treated with LPS and cordycepin extracts, their phagocytic activity increased dose-dependently (Figure 5a,b). Results of t-SNE show that most cells had a weak phagocytic activity, but when treated with 1 µg/mL cordycepin extract, macrophages showed a strong phagocytic activity (Figure 5c). When macrophages were treated with 0.1 µg/mL cordycepin extract, their phagocytic activity was dramatically stimulated compared to that of cells treated only with LPS. Moreover, when cells were treated with 1 µg/mL cordycepin extract, the phagocytic activity of macrophages increased 13 times compared to that observed following treatment with LPS only (Figure 5d).

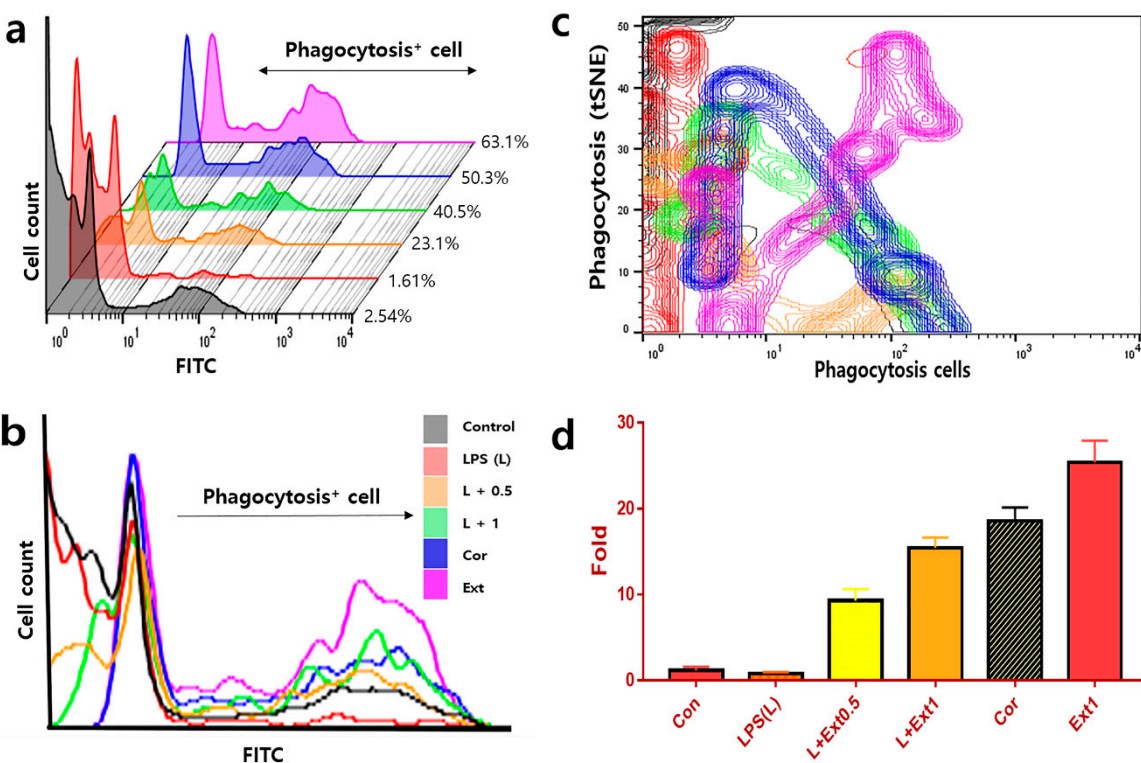

**Figure 5.** Activation of phagocytosis by *Cordyceps militaris* extracts in macrophages under LPS stress. Particles of *E. coli* were labeled with FITC. The phagocytosed macrophages were counted by flow cytometry and were analyzed by FlowJo software 10.6.1. The (**a**,**b**) (supplement graph for the panel (**a**)) panels show histograms of activation of phagocytosis by *Cordyceps militaris* extracts, and the (**c**) panel shows the visualizing datum using t-SNE for the phagocytosis. Panel (**d**) shows the relative fold changes for phagocytic activation by serial doses of the extract under LPS stress. LPS; lipopolysaccharide, Ext; extract, Cor; cordycepin ($p < 0.05$).

### 3.5. Differentiation Patterns of Monocytes and T Lymphocytes

Although Treatment with supernatant of LPS-challenged macrophages blocked the differentiation of CD11b$^+$ and Xcr1$^+$ cells from monocytes, followed by treatment with the conditioned medium from macrophages treated with cordycepin and LPS, dendritic cells (DCs) were strongly differentiated toward CD11b$^+$ and Xcr1$^+$ cells because their densities increased 13.6 times and 6.26 times, respectively, compared to those observed following treatment with conditioned medium from LPS- challenged macrophages (Figure 6). Compared to cordycepin, the extract conditioned medium slightly enhanced their differentiation from monocytes than differentiation by the cordycepin conditioned medium (Figure 6). The conditioned medium from LPS- challenged macrophages under cordycepin stimulated the differentiation of T cells to regulatory T cells (T$_{reg}$) and natural killer T-like (NKTL) cells (Figure 7). Notably, the differentiation of NKTL cells increased by 6.75 times compared to that observed following treatment with conditioned medium from macrophages challenged with only LPS (Figure 7c,d). Moreover, although T cells were stressed with supernatant of LPS-challenged macrophages, the differentiation of T$_{reg}$ and NKTL cells, followed by treatment with conditioned medium from cordycepin-treated macrophages, was increased about 1.67 times and 6.73 times, respectively, compared to that observed when treated with conditioned medium from LPS-challenged macrophages (Figure 7b,d). The left extract (0.001 µg/mL) in the conditioned medium not induced apoptosis of macrophages and did not affect alteration of cytokine levels (Figure S1).

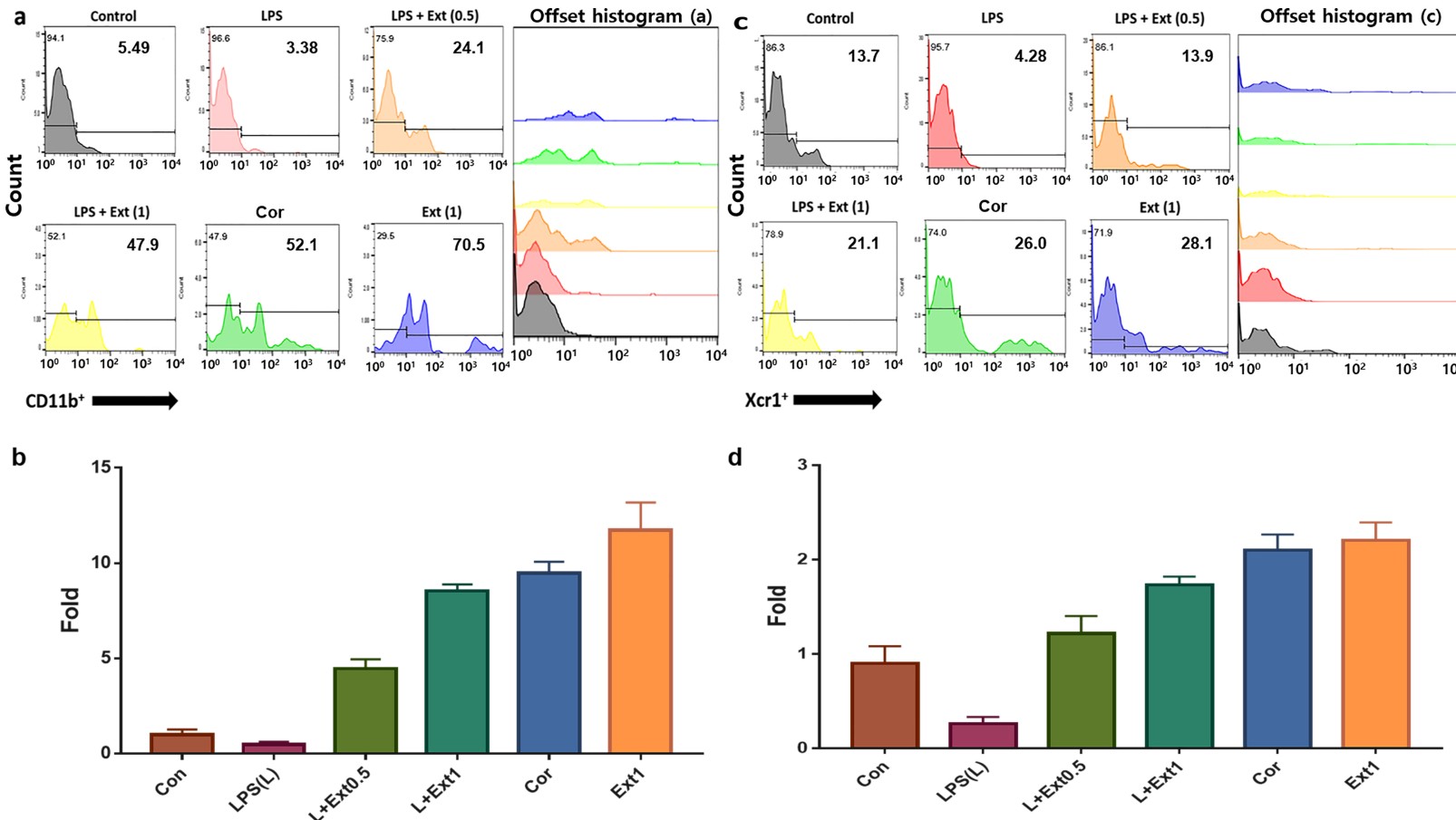

**Figure 6.** Dendritic differentiation by activated macrophages in the various conditioned media. Differentiated dendritic cells were counted by flow cytometry and were analyzed using the FlowJo software 10.6.1. Panels (**a**,**b**) show the density of CD11b[+] cells under various conditions as follows: LPS only, extract 0.5 μg/mL and 1 μg/mL (Ext 0.5, 1), followed by LPS treatment, cordycepin, and extract. Panels (**c**,**d**) show the density of Xcr1[+] cells under the same conditions. LPS; lipopolysaccharide, Ext; extract, Cor; cordycepin ($p < 0.05$).

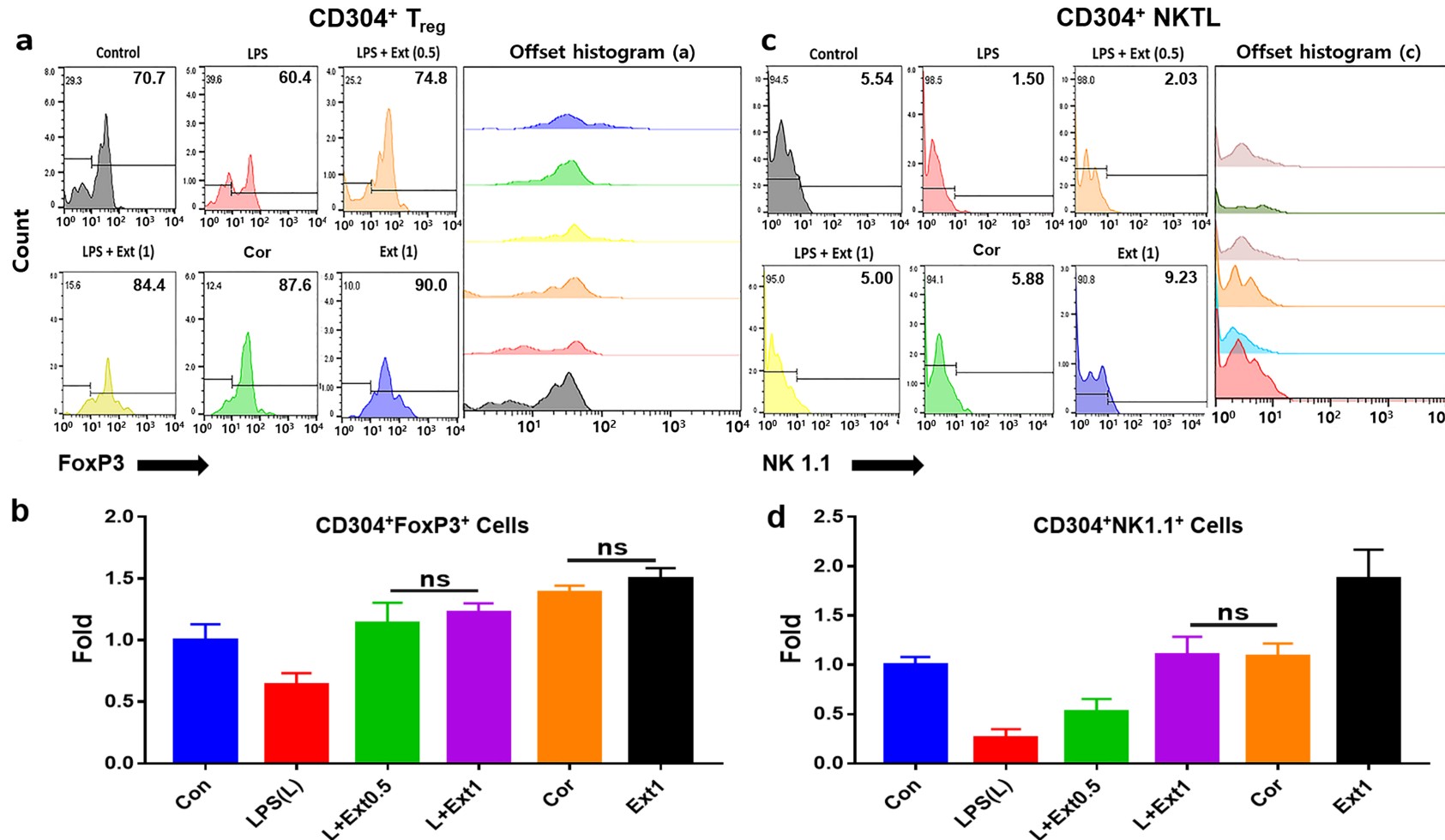

**Figure 7.** Lymphocyte differentiation by activated macrophages in the various conditioned media. After gating with CD304, the gated cells were distinguished from CD304+ $T_{reg}$ and natural killer T-like (NKTL) cells through expression of FoxP3 and NK1.1. Differentiated lymphocytes were counted by flowcytometry and were analyzed using the FlowJo software 10.6.1. Panels (**a**,**b**) show the density of CD304+FoxP3+ cells under various conditions as follows: LPS only, extract 0.5 µg/mL and 1 µg/mL (Ext 0.5, 1), followed by LPS treatment, cordycepin, and extract. Panels (**c**,**d**) show the density of CD304+NK1.1+ cells under the same conditions. LPS; lipopolysaccharide, Ext; extract, Cor; cordycepin, ns; not significant ($p < 0.05$).

## 4. Discussion

Although there are many research studies that documented immunological functions of cordycepin and *C. militaris* extract [7], they are poorly understood for immunological modulation by macrophage under the extract.

Cordyceps has been used as a healthy food because of its various effects of cleansing the blood, suppressing harmful bacteria and inflammation, and improving immunity [23]. Cordycepin, a functional substance of *Cordyceps sinensis*, was first found in *C. militaris* fruiting bodies [24]. In addition, 3′-deoxyadenosine has many biological activities, such as anti-inflammatory, antioxidative and anti-aging, antitumor, anti-cancer, and anti-leukemia activities [8]. LPS, one of the components of the cell wall of gram-negative bacteria, induces a strong inflammatory response through toll-like receptor 4 (TLR4) activation in host cells and sepsis. LPS is important for inflammatory reactions in host cells, and it plays a role in protecting bacterial cells from external toxic substances and antibiotics.

The total free amino acid, adenosine, and cordycepin contents were 83.35 mg/g, 0.24%, and 1.33%, respectively, in the *C. militaris* hydrolytic extract [25]. The concentration of cordycepin was higher than that of adenosine [25]. The concentrations of adenosine and 3′-deoxyadenosine in the extracts were found to be $0.05 \pm 0.01$ μg/mL and $0.47 \pm 0.09$ μg/mL, respectively, using HPLC. The cytotoxic concentrations of cordycepin in various cell lines ranged from 300 μM to 400 μM [26]. Although the extracts in this study contained low contents of cordycepin, the cytotoxic concentration ($CC_{50}$) of the extract was established at these low concentrations. This result means that other compounds excluding cordycepin affect attenuation of cellular viability in macrophages.

Recently, the relationship between inflammation and chronic disease has been reported. Although inflammation acts as a protective barrier against bacteria in the early stages of damage, it causes various chronic diseases such as obesity, diabetes, cancer, and brain and heart diseases [27]. Macrophages derived from monocyte precursors undergo specific differentiation depending on the local tissue environment. They respond to environmental cues within tissues such as cell damage, activated lymphocytes, or microbial products to differentiate into distinct functional phenotypes [28]. Macrophages are polarized into two phenotypes in response to various factors—classical macrophages M1 and alternative macrophages and M2 [15,26]. M1 macrophages are typically activated by IFN-γ or LPS and produce pro-inflammatory cytokines, IL-1β, and TNF-α, to kill pathogens. However, excessive production of M1 macrophages is associated with the onset of chronic inflammatory diseases [15]. M2 macrophages are selectively activated by exposure to specific cytokines such as IL-4, IL-10, or IL-13, and are involved in wound healing and tissue repair with anti-inflammatory activity [18,19]. M1 and M2 cells secrete major cytokines, and the process of the production of these cytokines has the histopathological characteristics of various inflammation-related diseases. Although the M1 phenotype is essential for the immune system, prolonged M1 polarization leads to several diseases [29]. The M2 phenotype, on the other hand, attenuates inflammation through phagocytosis of a pathogen [15,30]. In this study, the extracts dose-dependently suppressed the M1 polarization of macrophages but activated M2 polarization. Interestingly, qPCR showed that the levels of TNF-α and IL-1β were notably downregulated in the macrophages at 1 μg/mL of the extracts (Figure 2). Furthermore, unlike LPS altered cytokine profile such as increasing IL-18 and decreasing of IL-4 and IL-13 in macrophages, the extract enhanced increasing of IL-4 and 13 (Figure S1). Various compounds in phytoextracts such as melatonin and L-3-n-butylphthalide from celery and polyphenols from cocoa have been shown to activate the M2 polarization of macrophages [31–33]. Flow-cytometry and qPCR analysis in this research showed that the extracts significantly inhibited TNF-α and IL-1β by suppression of M1 polarization in the macrophages (Figures 3 and 4). These results also suggest that the extracts strongly induced suppression of pro-inflammatory cytokines and activation of phagocytosis and contained compounds that activated M2 polarization.

Macrophages also showed enhanced phagocytic activity dose-dependently. As per the t-SNE results, macrophages exposed to 1 μg/mL of the extracts formed populations

with strong activity (Figure 5c). This result suggests that the extracts caused the activation of M2 polarization and phagocytic activity of M2 macrophages.

Activated macrophages modulated the differentiation of dendritic and T cells. First, activated macrophages strongly increased the number of CD11b$^+$ and Xcr1$^+$ DCs (Figure 5). From some reports [34–36], pre-conventional dendritic cells (pre-cDC) among DC precursors were differentiated by interferon regulatory factor 4 (IRF4), macrophage colony-stimulating factor receptor (M-CSFR), and granulocyte–macrophage colony-stimulating factor receptor (GM–CSF). In addition, IRF4 induces downregulation of pro-inflammatory cytokines in leukocytes [36]. Xcr1$^+$ DCs enhance T-cell survival and activation [37], while CD11b$^+$ DCs are involved in antigen presentation to CD4$^+$ cells associated with the induction of Th2 or Th17 cells [35,38]. Furthermore, unlike LPS altered profile of inflammatory cytokines as increasing IL-18 and decreasing of IL-4 and IL-13 in macrophages, the extract enhanced the increase of IL-4 and 13 (Figure S1). These results suggest that the extract activates macrophages to synthesize suppressors of pro-inflammatory signals and enhances DC differentiation.

In addition to stimulating CD304$^+$FoxP3$^+$ T$_{reg}$ cells, the number of CD304$^+$NK1.1$^+$ NKTL cells following incubation with conditioned medium from macrophages treated with the extract and LPS was 6.73 times higher than that observed when conditioned medium from LPS-challenged macrophages was used. FoxP3$^+$ T$_{reg}$ cells are known as homeostatic modulators of the immune system [39]. T$_{reg}$ cells play important roles in various immune systems, including hypersensitivity suppression, pathogenic immunopathology, and autoimmune disease suppression [40]. These results show that the extract activates macrophages to enhance the differentiation of CD304$^+$NK1.1$^+$ NKTL cells and suppress the inflammatory response induced by LPS. Although the *C. militaris* extract 1 μg/mL contained lower quantities of cordycepin than did the conditional medium from cordycepin-treated macrophages, cell differentiation was enhanced 1.6 times by the extract conditioned medium compared to the cordycepin conditioned medium. Additionally, in the results for comparison of CD3/CD28 and the extract stimulation (Figure S7), the stimulating capacity was evaluated similarly between them. These results suggest that the extract contained other bioactive compounds in addition to cordycepin that stimulated the differentiation of immune cells. Moreover, the extract is more effective to activate an immune response when compared to cordycepin.

With a mouse's cell line, this research documented that macrophages are an immune conductor under the extract. However, in vivo study for the extract will be considered to prove the activation of macrophage as an immune conductor by the extract. If results from in vivo research correspond to the results of this research, the extract is applied to a bioactive material in functional foods, cosmetic, and immunological additive for stock feeds.

## 5. Conclusions

Consequently, the *C. militaris* hydrolytic extract aids in the prevention and resolution of inflammation under LPS stress through four functions. First, the extract modulates macrophage polarization to achieve prolonged production of macrophages with the M2 phenotype. Second, the extract suppresses the secretion of pro-inflammatory cytokines from macrophages. Third, the extract enhances the phagocytic activity of macrophages. Fourth, the extract enhances the differentiation of DCs, T$_{reg}$ cells, and NKTL cells for the suppression of inflammation. Therefore, the *C. militaris* hydrolytic extract has strong effects on the modulation of immune actors, such as macrophages and dendritic cells, under inflammatory stress.

**Supplementary Materials:** The following are available online at https://www.mdpi.com/2076-3417/11/5/2206/s1. File S1; Figures S2 and S7: full gels for qPCR, Figure S1: results for ELISA and Figures S4–S6: flow cytometer and Figure S3: MTT assay, Table S1: list of primers for qPCR.

**Author Contributions:** Conceptualization and methodology, Y.P. and S.C.; writing—original draft preparation, Y.P.; writing—review and editing, B.K.; supervision, B.K. and S.G.L. All authors have read and agreed to the published version of the manuscript.

**Funding:** This research received no external funding.

**Institutional Review Board Statement:** Not applicable.

**Informed Consent Statement:** Not applicable.

**Data Availability Statement:** Not applicable.

**Acknowledgments:** This study was supported by LIFE TOGETHER, MITOS BIO and Korea University.

**Conflicts of Interest:** The authors declare no conflict of interest.

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
