# Peer review of "Effects of Cordyceps militaris Extracts on Macrophage as Immune Conductors"

_applsci, doi:10.3390/app11052206_

Round 1
Reviewer 1 Report
The manuscript was prepared very well. The introduction section justifies the purpose of the study. I congratulate the authors for the preparation of the manuscript
However, I have the following comments:
Introduction
Line 35: What species of Cordyceps produce Corycepin?
Line 39: please include this information. In addition to the therapeutic effects, have any side effects been reported? What doses are considered effective and which are toxic to produce these effects?
Line 49: Have previous manuscripts reported immunostimulatory or immunoregulatory functions in the inflammatory process?
Materials and Methods
The methodology is correct and very well described. Congratulations to the authors for this section, it is one of the strengths of the study
Results
The results are excellently described and meet the objectives set. The figures are representative of the main findings and contain critical information, but in general the titles should be improved and figure captions should be added with content that explains what is represented in the figure.
Figure 1: the figure captions are somewhat confusing. You must make a figure caption with the steps developed and the acronyms used. Please improve this aspect
Figure 2: In the two HPLC figures, the same is reprinted? please explain in each figure (a and b) what each figure contains to facilitate understanding of the manuscript
Figure 3 and 4: they are too small, and the title of the figure is unbreakable. Please put a figure caption on each one to improve the explanation.
Discussion
Could you include a small paragraph at the beginning of the discussion of the most relevant findings?
Line 232: add reference to the end of this paragraph
line 241 why could it affect cellular apoptosis? Briefly explain
lines 265-268: this paragraph contains some result that can be observed in some figure of your results? In the same way that they indicate in line 263
Please include a list of limitations of the study
Could a section of practical applications be made?

Author Response
I appreciate for improvement of our manuscript through your comments.
Comment 1: Line 35: What species of Cordyceps produce Corycepin?
Answer 1: We revised the sentence, Line 41
Comment 2: Line 39: please include this information. In addition to the therapeutic effects, have any side effects been reported? What doses are considered effective and which are toxic to produce these effects?
Answer 2: We added the sentence at line 45 as following;
Although cordycepin has various therapeutic effects including inhibition of mRNA polyadenylation and mRNA elongation, activation of AMPK and protein phosphatase [10-12], there are side effects of Cordyceps including increasing of autoimmune diseases, slow blood clotting and interaction with other drugs [13]. Suggesting dose of Cordyceps is two and three times with 1050mg extract or 0.14% adenosine [13].
Comment 3: Line 49: Have previous manuscripts reported immunostimulatory or immunoregulatory functions in the inflammatory process?
We revised the sentence as following;
Answer 3: Although reported researches have shown the anti-inflammatory effects of C. militaris extracts
Comment 4: Figure 1: the figure captions are somewhat confusing. You must make a figure caption with the steps developed and the acronyms used. Please improve this aspect
Answer 4: Under the extract, the supernatants derived from macrophages were collected and exposed to T cells and monocytes. M1 and 2; macrophage 1 and 2, DC; dendritic cells, NKT cells; natural killer cells
Comment 5: Figure 2: In the two HPLC figures, the same is reprinted? please explain in each figure (a and b) what each figure contains to facilitate understanding of the manuscript
Answer 5: We revised the sentences in Figure 2’s legend and manuscript as follows;
Legend : Panel a showed the overlay chromatogram of various concentrations of the cordycepin and adenosine standards. Panel b showed the concentration of cordycepin in Cordyceps militaris extracts.
Manuscript : Cordycepin and adenosine levels were quantified by HPLC using appropriate standards (Figure 2a)
Comment 6: Figure 3 and 4: they are too small, and the title of the figure is unbreakable. Please put a figure caption on each one to improve the explanation.
Answer 6: We revised Figure 3 and 4
Discussion
Comment 7: Could you include a small paragraph at the beginning of the discussion of the most relevant findings?
Answer 7: We added the sentence at Discussion as following;
Although there are many researches documented immunological functions of cordycepin and C. militaris extract [21], there are poorly understood for immunological modulation by macrophage under the extract.
Comment 8: Line 232: add reference to the end of this paragraph
Answer 8: we added the reference
Comment 9: line 241 why could it affect cellular apoptosis? Briefly explain
Answer 9: We revised the sentence as following;
Although the extracts in this study contained low contents of cordycepin, the cytotoxic concentration (CC50)of the extract was established at these low concentrations. This result means that other compounds excluding cordycepin affect attenuation of cellular viability in macrophages.
Comment 10: lines 265-268: this paragraph contains some result that can be observed in some figure of your results? In the same way that they indicate in line 263
Answer 10: We revised the sentence
Comment 11: Please include a list of limitations of the study. Could a section of practical applications be made?
Answer 11: We added the sentence at Discussion as follows;
In a cell level, this research documented that macrophages are an immune conductor under the extract. However, in vivo study for the extract will be considered to prove the activation of macrophage as an immune conductor by the extract. If results from in vivo research correspond to the results of this research, the extract is applied to a functional material in functional foods, cosmetic and immunological additive for stock feeds.

Reviewer 2 Report
The manuscript „Effects of Cordyceps militaris extracts on macrophage as immune conductor” by Choi et al. covers an interesting topic. The authors try to elucidate the impact of C. militaris extract on macrophages and the subsequent effects of soluble factors produced by such macrophages on other immune cells.
Although the topic is interesting, the manuscript raises several points that need to be addressed.
Major points:
- Introduction line 36 f: The citations are wrong, only one paper of references 5-8 contains the cited information. Furthermore, please do not over-interpret the data because most was shown only in vitro
- Line 41: “Macrophages… act as anti-inflammatory antigen-presenting cells” is a wrong statement. They can, but most of the time they are pro-inflammatory
- Line 42: “They drive homeostasis…” What kind of?
- Line 48: again wrong citations 5-8
- Materials and methods 2.3: Here it is stated that T cells and monocytes were only treated with the macrophage conditioned medium, in line 219 it is stated that they were also treated with LPS. What is correct? If they were treated with LPS, what was the exact protocol?
- The authors chose cellular models that are not established to study immune regulation. For example, the polarization of a T cell line towards NKT or Treg cells by LPS is no physiological setting. T cells cannot be activated by LPS alone; I would suggest to follow established protocols by activating the cells with anti-CD3 or anti-TCR antibodies in combination with anti-CD28
- Materials and Methods 2.6: “After treatment with LPS…” How long? What concentration?
- The authors determine protein expression only on mRNA level (with two different methods). They should verify their data by protein levels (i.e. ELISA)
- Line 111: Were the three experiments independent? An ANOVA can only be performed on normally distributed data which is hard to test on n=3
- Line 129/130: Please provide proof that the TK-1 cell line resembles immature thymocytes. If it doesn’t, the development to NKT cells is not possible
- Most important point for the interpretation of all following data: Please do a toxicity assessment on the extract. Most of the results can be explained by dying/apoptotic cells. The authors state in line 241 that other compounds could affect apoptosis. So for every experimental setup please make sure that all cells are in the same condition. The FACS plots for figure 5 suggest that the cells might be dead. The authors could easily provide information on FSC/SSC (percentage of cells in live gate)
- Most of the effects on monocytes and macrophages can be explained by different cytokine levels in the supernatant, only secondary to the effect of the extract
- Figure 4: Polarization patterns of macrophages. The data shown is generated by FISH. Please provide information how the detection of M1 and M2 was performed since it is not stated in the material and methods section. Please also provide percentages of the populations of the total cell count. Again, the FACS patterns suggest that many cells were dead at the time of analysis
- Figure 5 (the first one): LPS should induce phagocytosis. However, LPS is also present on the E. coli that are used for the experiment. Maybe the pre-treatment with LPS lead to an exhaustion of the cells? Again, a live/dead analysis would help to understand the results, because the autofluorescence of dying cells could account for the positive staining after treatment
- Figure 5 (the second): Please provide FCS/SSC, the panels suggest that there were not many cells left; Please explain what the right panels in a and b show
- Line 211-214: LPS treatment of T cells does neither induce NKT nor Tregs
- Figure 7a: The control already has >70% Tregs what is unphysiologically high (normal rate is 5%). Please provide your gating strategy and back up this data with primary T cells
Minor points:
- The language should be edited
- Abstract: In the abstract, the authors compare cells “under the extract” with LPS-treated cells. However, also the first ones were LPS treated
- Materials and methods 2.3: were the cells kept with LPS and additionally treated with the extract? Or was the LPS washed away?
- Materials and methods line 95: Please provide the sequence of the probes
- Materials and methods line 100 and 108: FlowJo is a software by TreeStar, not BD
- Materials and methods line 96, 100 and 108: The company’s name is BD Biosciences
- Materials and methods line 102-103: please provide information on the clones
- Materials and methods line 113: The company’s name is GraphPad
- Materials and methods line 113: Please provide information in the asterisks used
- Figure legends: Provide information on the asterisks (**missing in figure 3, no asterisks at all in 1, 2, 4, 5, 6 and 7; does this mean all bars except the “n.s.” ones are significant?)
- There are two figure 5 (phagocytosis and dendritic cells)
- First figure 5: except for the design, what is the difference between 5 a and b?
- Interpretation of the data: If only the supernatant of macrophages is used for the treatment of monocytes and T cells: How much of the extract is still left in the supernatant? Furthermore, if the macrophages die under the treatment, less/other cytokines are produced by them and hence many effects can be explained by different cytokine levels
Author Response
I appreciate for improvement of our manuscript through your comments.
Comment 1: Introduction line 36 f: The citations are wrong, only one paper of references 5-8 contains the cited information. Furthermore, please do not over-interpret the data because most was shown only in vitro
Answer 1: we revised the references
Comment 2: Line 41: “Macrophages… act as anti-inflammatory antigen-presenting cells” is a wrong statement. They can, but most of the time they are pro-inflammatory
Answer 2: We revised the sentence as following
Macrophages, derived from monocytes, play an important role in phagocytosis in disease lesions and act as anti-inflammatory cells
Comment 3: Line 42: “They drive homeostasis…” What kind of?
Answer 3: We revised the sentence as following
They also involve in modulation of innate and acquired immunities
Comment 4: Line 48: again wrong citations 5-8
Answer 4: We revised the references
Comment 5: Materials and methods 2.3: Here it is stated that T cells and monocytes were only treated with the macrophage conditioned medium, in line 219 it is stated that they were also treated with LPS. What is correct? If they were treated with LPS, what was the exact protocol?
Answer 4: We revised the references as follows
Cells were exposed to four conditions including LPS (100 ng/mL), the extracts (Ext), cordycepin (Cor, 30 μM) (Sigma), pre-LPS and post-extract (LPS+Ext). Cells were exposed to LPS for 12 h, and subsequently treated with C. militaris extracts (0.1, 0.25, 0.5, or 1 µg/mL) for 8 h. After exposing, the supernatants from the four conditions were collected. To analyze cellular differentiation, monocytes (JAWS ǁ; ATCC CRL-11904) and T cells (TK-1; ATCC CRL-2396) were cultured with the four conditioned mediums derived from the supernatant of macrophages treated with Ext1(extract 1 µg/mL), LPS, LPS+Ext1 or 0.5 and Cor 30 μM.
Comment 5: The authors chose cellular models that are not established to study immune regulation. For example, the polarization of a T cell line towards NKT or Treg cells by LPS is no physiological setting. T cells cannot be activated by LPS alone; I would suggest to follow established protocols by activating the cells with anti-CD3 or anti-TCR antibodies in combination with anti-CD28
Answer 5: We revised the references as following
To analyze cellular differentiation, monocytes (JAWS ǁ; ATCC CRL-11904) and T cells (TK-1; ATCC CRL-2396) were cultured with the four conditioned mediums derived from the supernatant of macrophages treated with Ext1(extract 1 µg/mL), LPS, LPS+Ext1 or 0.5 and Cor 30 μM.
Comment 6: Materials and Methods 2.6: “After treatment with LPS…” How long? What concentration?
Answer 6: We revised the references as following
After treatment with LPS for 12h and C. militaris extracts, macrophages were exposed to fluorescence-labelled E. coli particles for 8h and phagocytosis was examined using Vybrant™ Phagocytosis Assay Kit (Thermo Fisher scientific Inc, Germering, Germany) for 2 h.
Comment 7: The authors determine protein expression only on mRNA level (with two different methods). They should verify their data by protein levels (i.e. ELISA)
Comments 7: We added the supplementary file(S2) containing the results of ELISA and qPCR
Comment 8: Line 111: Were the three experiments independent? An ANOVA can only be performed on normally distributed data which is hard to test on n=3
Answer 8: we did three independent experiments and did estimated 5 times per one experiment. Furthermore, a sample without normality was analyzed using non-parametric test ( Kruskal-wallis H Test). We revised the sentence as follows;
All experiments were three independent experiments with estimating 5 times per one experiment (n = 3), and the data were analyzed by one-way analysis of variance (ANOVA) with Post Hoc test (Scheffe’s method) and a sample without normality was analyzed using non-parametric test ( Kruskal-wallis H Test) with the SPSS v26 (IBM, NY, USA) and Prism 7 (GrapgPad, CA, USA) softwares.
Comment 9: Line 129/130: Please provide proof that the TK-1 cell line resembles immature thymocytes. If it doesn’t, the development to NKT cells is not possible
Answer 9: We revised as natural killer T like cells (NKTLC)
Comment 10: Most important point for the interpretation of all following data: Please do a toxicity assessment on the extract. Most of the results can be explained by dying/apoptotic cells. The authors state in line 241 that other compounds could affect apoptosis. So for every experimental setup please make sure that all cells are in the same condition.
Answer 10: We revised the sentence as follows;
Although the extracts in this study contained low contents of cordycepin, the cytotoxic concentration (CC50) of the extract was established at these low concentrations. This result means that other compounds excluding cordycepin affect attenuation of cellular viability in macrophages.
Comment 11: The FACS plots for figure 5 suggest that the cells might be dead. The authors could easily provide information on FSC/SSC (percentage of cells in live gate)
Answer 11: We revised Figure 5
Comment 12: Most of the effects on monocytes and macrophages can be explained by different cytokine levels in the supernatant, only secondary to the effect of the extract
Answer 12: We added the supplementary file(S2) containing the results of ELISA and qPCR
Comment 13: Figure 4: Polarization patterns of macrophages. The data shown is generated by FISH. Please provide information how the detection of M1 and M2 was performed since it is not stated in the material and methods section. Please also provide percentages of the populations of the total cell count. Again, the FACS patterns suggest that many cells were dead at the time of analysis
Answer 13: The information for fluorescence labeled probes was already described at 2.5 section and we revised Figure 4
Comment 14: Figure 5 (the first one): LPS should induce phagocytosis. However, LPS is also present on the E. coli that are used for the experiment. Maybe the pre-treatment with LPS lead to an exhaustion of the cells? Again, a live/dead analysis would help to understand the results, because the autofluorescence of dying cells could account for the positive staining after treatment
Answer 14: Incorrect data were reflected to graph. We revised Figure 5
Comment 15: Figure 5 (the second): Please provide FCS/SSC, the panels suggest that there were not many cells left; Please explain what the right panels in a and b show
Answer 15: We revised Figure 5
Comment 16: Line 211-214: LPS treatment of T cells does neither induce NKT nor Tregs
Answer 16: In this research, T cells and monocytes were exposed to various conditioned media. We revised the sentence as following;
Moreover, although T cells were stressed with conditioned medium by LPS, the differentiation of Treg and NKT cells following treatment with conditioned medium from cordycepin-treated macrophages was increased about 1.67 and 6.73 times, respectively compared to that observed when treated with conditioned medium from LPS-stimulated macrophages
Comment 17: Figure 7a: The control already has >70% Tregs what is unphysiologically high (normal rate is 5%). Please provide your gating strategy and back up this data with primary T cells
Answer 17: After gating with FoxP3 or NK1.1, CD304+ Treg and NKT cells were evaluated.
We described information for the gating in legend of Figure 7
Minor points:
Comment 18: The language should be edited
Answer 18: After review, we will use English editing service of MDPI or Editage
Comment 19: Abstract: In the abstract, the authors compare cells “under the extract” with LPS-treated cells. However, also the first ones were LPS treated
Answer 19: This research was involved in four condition including non-treating, LPS treating, After LPS treating then exposing to the extract, exposing to the extract. The referred sentence mean after LPS treating then exposing to the extract.
Comment 20: Materials and methods 2.3: were the cells kept with LPS and additionally treated with the extract? Or was the LPS washed away?
Answer 20: We revised the sentence as following;
Cells were exposed to LPS for 12 h and cleaned with PBS (pH 7.4). Subsequently, the cleaned cells were treated with C. militaris extracts (0.1, 0.25, 0.5, or 1 µg/mL) for 8 h.
Comment 21: Materials and methods line 95: Please provide the sequence of the probes
Answer 21: We added the segurences at 2.5
Comment 22: Materials and methods line 100 and 108: FlowJo is a software by TreeStar, not BD
Answer 22: Recent, BD has acquired TreeStar
Comment 23: Materials and methods line 96, 100 and 108: The company’s name is BD Biosciences
Answer 23: We revised all
Comment 24: Materials and methods line 102-103: please provide information on the clones
Answer 24: We added information of the clones
Comment 25: Materials and methods line 113: The company’s name is GraphPad
Answer 25: We revised the text
Comment 26: Materials and methods line 113: Please provide information in the asterisks used
Answer 26: Line 113 is the title for table1 but I couldn’t found the asterisks
Answer 27: Figure legends: Provide information on the asterisks (**missing in figure 3, no asterisks at all in 1, 2, 4, 5, 6 and 7; does this mean all bars except the “n.s.” ones are significant?)
Answer 27: we revised the legends
Comment 28: There are two figure 5 (phagocytosis and dendritic cells)
Answer 28: We revised the number
Comment 29: First figure 5: except for the design, what is the difference between 5 a and b?
Answer 29: The 5b, two dimensional graph is supplement for 5a to understand difference of the counts among the groups. We described the mean of the graph 5b in the legend.
Comment 30: Interpretation of the data: If only the supernatant of macrophages is used for the treatment of monocytes and T cells: How much of the extract is still left in the supernatant? Furthermore, if the macrophages die under the treatment, less/other cytokines are produced by them and hence many effects can be explained by different cytokine levels
Answer 30:
To evaluate effects of secretory factors derived from macrophages, we treated 10ul of the supernatant to monocytes and T cells respectively and the results were compensated with results for directly treating with the same as the concentration of the extract in the treated supernatant. We added the sentence at 2.3

Reviewer 3 Report
This is a very interesting essay regarding the newest aspects of Cordiceps. It is very adequately planned and possesses quite an interesting scheme. It shows a design which is coherent with the results. I do have nonetheless some suggestions which could improve the project.
To begin wif th, iNOS should be studied, as much as COX2 is, since both possess a notorious importance in the inflammatory processes.
In the same way, another key component in inflammation which is the PPARgamma expression should also be analyzed.
As a last suggestion, there d be an interest in the study of the different subunits of NF-κB, as well as the redox status.
In my point of view, these are all elements which should be included in the study, in order to notoriously enhance the relevance of this essay.
Thanks.
Author Response
I appreciate for improvement of our manuscript through your comments.
Comment 1: To begin with, iNOS should be studied, as much as COX2 is, since both possess a notorious importance in the inflammatory processes. In the same way, another key component in inflammation which is the PPARgamma expression should also be analyzed. As a last suggestion, there d be an interest in the study of the different subunits of NF-κB, as well as the redox status.
Answer 1: We added the supplementary file(S2) containing the results of qPCR for iNOS, COX2, PPARgamma and NF-κB

Round 2
Reviewer 2 Report
The authors greatly improved the manuscript “Effects of Cordiceps militaris extracts on macrophage as immune conductor”. They included new data on the protein expression of the macrophages. However, some questions remained unanswered:
Comment 2: Line 41: “Macrophages… act as anti-inflammatory antigen-presenting cells” is a wrong statement. They can, but most of the time they are pro-inflammatory
Answer 2: We revised the sentence as following
Macrophages, derived from monocytes, play an important role in phagocytosis in disease lesions and act as anti-inflammatory cells
- The authors should include that macrophages also act as pro-inflammatory cells (depending on their polarization)
Comment 5: Materials and methods 2.3: Here it is stated that T cells and monocytes were only treated with the macrophage conditioned medium, in line 219 it is stated that they were also treated with LPS. What is correct? If they were treated with LPS, what was the exact protocol?
Answer 4: We revised the references as follows
Cells were exposed to four conditions including LPS (100 ng/mL), the extracts (Ext), cordycepin (Cor, 30 μM) (Sigma), pre-LPS and post-extract (LPS+Ext). Cells were exposed to LPS for 12 h, and subsequently treated with C. militaris extracts (0.1, 0.25, 0.5, or 1 µg/mL) for 8 h. After exposing, the supernatants from the four conditions were collected. To analyze cellular differentiation, monocytes (JAWS ǁ; ATCC CRL-11904) and T cells (TK-1; ATCC CRL-2396) were cultured with the four conditioned mediums derived from the supernatant of macrophages treated with Ext1(extract 1 µg/mL), LPS, LPS+Ext1 or 0.5 and Cor 30 μM.
- Thank you for the clarification. Please also revise the respective text sections (for example line 239: “…LPS treatment blocked the differentiation…” If I understand this right, it should be rephrased to “Treatment with supernatant of LPS-challenged macrophages”
Comment 5: The authors chose cellular models that are not established to study immune regulation. For example, the polarization of a T cell line towards NKT or Treg cells by LPS is no physiological setting. T cells cannot be activated by LPS alone; I would suggest to follow established protocols by activating the cells with anti-CD3 or anti-TCR antibodies in combination with anti-CD28
Answer 5: We revised the references as following
To analyze cellular differentiation, monocytes (JAWS ǁ; ATCC CRL-11904) and T cells (TK-1; ATCC CRL-2396) were cultured with the four conditioned mediums derived from the supernatant of macrophages treated with Ext1(extract 1 µg/mL), LPS, LPS+Ext1 or 0.5 and Cor 30 μM.
- So the T cells have not been activated in any way but only incubated in the presence of the supernatants? It would be interesting to check what would happen during stimulation (CD3/CD28)
Comment 10: Most important point for the interpretation of all following data: Please do a toxicity assessment on the extract. Most of the results can be explained by dying/apoptotic cells. The authors state in line 241 that other compounds could affect apoptosis. So for every experimental setup please make sure that all cells are in the same condition.
Answer 10: We revised the sentence as follows;
Although the extracts in this study contained low contents of cordycepin, the cytotoxic concentration (CC50) of the extract was established at these low concentrations. This result means that other compounds excluding cordycepin affect attenuation of cellular viability in macrophages.
- Please include data on cell viability for all the experiments (for example percent of live cells in FSC/SSC of PI staining). Different cells respond differently.
Comment 11: The FACS plots for figure 5 suggest that the cells might be dead. The authors could easily provide information on FSC/SSC (percentage of cells in live gate)
Answer 11: We revised Figure 5
- No new information on live/dead was added
Comment 12: Most of the effects on monocytes and macrophages can be explained by different cytokine levels in the supernatant, only secondary to the effect of the extract
Answer 12: We added the supplementary file(S2) containing the results of ELISA and qPCR
- Please discuss that the altered cytokine profile can explain most of the results (=indirect effect of the extract)
Comment 13: Figure 4: Polarization patterns of macrophages. The data shown is generated by FISH. Please provide information how the detection of M1 and M2 was performed since it is not stated in the material and methods section. Please also provide percentages of the populations of the total cell count. Again, the FACS patterns suggest that many cells were dead at the time of analysis
Answer 13: The information for fluorescence labeled probes was already described at 2.5 section and we revised Figure 4
- No new information on live/dead was added
Comment 14: Figure 5 (the first one): LPS should induce phagocytosis. However, LPS is also present on the E. coli that are used for the experiment. Maybe the pre-treatment with LPS lead to an exhaustion of the cells? Again, a live/dead analysis would help to understand the results, because the autofluorescence of dying cells could account for the positive staining after treatment
Answer 14: Incorrect data were reflected to graph. We revised Figure 5
- Now, the phagocytosis of LPS-treated cells is even lower than in the first version although it should potently induce phagocytosis.
- No new information on live/dead was added what would be important to exclude autofluorescence
Comment 15: Figure 5 (the second): Please provide FCS/SSC, the panels suggest that there were not many cells left; Please explain what the right panels in a and b show
Answer 15: We revised Figure 5
- No new information on live/dead was added (now figure 6)
- No information was added what the right panels in a) and c) show
Comment 17: Figure 7a: The control already has >70% Tregs what is unphysiologically high (normal rate is 5%). Please provide your gating strategy and back up this data with primary T cells
Answer 17: After gating with FoxP3 or NK1.1, CD304+ Treg and NKT cells were evaluated.
We described information for the gating in legend of Figure 7
- Please provide FACS plots for the gating strategy
- It would be nice to use primary T cells to evaluate physiological effects.
Minor points:
Comment 24: Materials and methods line 102-103: please provide information on the clones
Answer 24: We added information of the clones
- Please provide the exact name of the clones, not only the species (or were they polyclonal?)
Comment 26: Materials and methods line 113: Please provide information in the asterisks used
Answer 26: Line 113 is the title for table1 but I couldn’t found the asterisks
- The asterisks are in figure 3
Answer 27: Figure legends: Provide information on the asterisks (**missing in figure 3, no asterisks at all in 1, 2, 4, 5, 6 and 7; does this mean all bars except the “n.s.” ones are significant?)
Answer 27: we revised the legends
- Unfortunately, I still don’t understand. Does it mean that each and every bar is significantly different from all the others? Why did you chose a different way of labeling for figure 3?
Comment 30: Interpretation of the data: If only the supernatant of macrophages is used for the treatment of monocytes and T cells: How much of the extract is still left in the supernatant? Furthermore, if the macrophages die under the treatment, less/other cytokines are produced by them and hence many effects can be explained by different cytokine levels
Answer 30:
To evaluate effects of secretory factors derived from macrophages, we treated 10ul of the supernatant to monocytes and T cells respectively and the results were compensated with results for directly treating with the same as the concentration of the extract in the treated supernatant. We added the sentence at 2.3
- Does the extract itself have any effect on the cells? If so, what effects were found? If not, the information is still important and should be discussed.
Author Response
I appreciate for your detail comments. With your comments, our manuscript was improved and clarified to explain goals in this research.
Comment 2: Line 41: “Macrophages… act as anti-inflammatory antigen-presenting cells” is a wrong statement. They can, but most of the time they are pro-inflammatory
Answer 2: We revised the sentence as following
Macrophages, derived from monocytes, play an important role in phagocytosis in disease lesions and act as anti-inflammatory cells
- The authors should include that macrophages also act as pro-inflammatory cells (depending on their polarization)
Answer 2’: We revised the sentence as following
Macrophages, derived from monocytes, play an important role in phagocytosis in disease lesions and act as anti-inflammatory and pro-inflammatory cells depending on their polarization.
Comment 4: Materials and methods 2.3: Here it is stated that T cells and monocytes were only treated with the macrophage conditioned medium, in line 219 it is stated that they were also treated with LPS. What is correct? If they were treated with LPS, what was the exact protocol?
Answer 4: We revised the references as follows
Cells were exposed to four conditions including LPS (100 ng/mL), the extracts (Ext), cordycepin (Cor, 30 μM) (Sigma), pre-LPS and post-extract (LPS+Ext). Cells were exposed to LPS for 12 h, and subsequently treated with C. militaris extracts (0.1, 0.25, 0.5, or 1 µg/mL) for 8 h. After exposing, the supernatants from the four conditions were collected. To analyze cellular differentiation, monocytes (JAWS ǁ; ATCC CRL-11904) and T cells (TK-1; ATCC CRL-2396) were cultured with the four conditioned mediums derived from the supernatant of macrophages treated with Ext1(extract 1 µg/mL), LPS, LPS+Ext1 or 0.5 and Cor 30 μM.
- Thank you for the clarification. Please also revise the respective text sections (for example line 239: “…LPS treatment blocked the differentiation…” If I understand this right, it should be rephrased to “Treatment with supernatant of LPS-challenged macrophages”
Answer 4’: Thanks for suggesting of your excellent phrase. We revised all sentences with your phrase.
Comment 5: The authors chose cellular models that are not established to study immune regulation. For example, the polarization of a T cell line towards NKT or Treg cells by LPS is no physiological setting. T cells cannot be activated by LPS alone; I would suggest to follow established protocols by activating the cells with anti-CD3 or anti-TCR antibodies in combination with anti-CD28
Answer 5: We revised the references as following
To analyze cellular differentiation, monocytes (JAWS ǁ; ATCC CRL-11904) and T cells (TK-1; ATCC CRL-2396) were cultured with the four conditioned media derived from the supernatant of macrophages treated with Ext1(extract 1 µg/mL), LPS, LPS+Ext1 or 0.5 and Cor 30 μM.
- So the T cells have not been activated in any way but only incubated in the presence of the supernatants? It would be interesting to check what would happen during stimulation (CD3/CD28)
Answer 5’: We added the data at supplementary file (S1) and revised the sections; materials and methods and discussion; line 100 and 351.
Comment 10: Most important point for the interpretation of all following data: Please do a toxicity assessment on the extract. Most of the results can be explained by dying/apoptotic cells. The authors state in line 241 that other compounds could affect apoptosis. So for every experimental setup please make sure that all cells are in the same condition.
Answer 10: We revised the sentence as follows;
Although the extracts in this study contained low contents of cordycepin, the cytotoxic concentration (CC50) of the extract was established at these low concentrations. This result means that other compounds excluding cordycepin affect attenuation of cellular viability in macrophages.
- Please include data on cell viability for all the experiments (for example percent of live cells in FSC/SSC of PI staining). Different cells respond differently.
Answer 10’: We added the data (MTT assay) at supplementary file (S1).
Comment 11: The FACS plots for figure 5 suggest that the cells might be dead. The authors could easily provide information on FSC/SSC (percentage of cells in live gate)
Answer 11: We revised Figure 5
- No new information on live/dead was added
Answer 11’: We added the data (FSC/SSC) at supplementary file (S1)
To evaluate phagocytic activity of macrophages, we treated the FITC labeled particles of E. coli in macrophages. The phagocyted macrophages contain the FITC-labeled particles in their cytoplasm. Populations in below 101 were unphagocytosed macrophages. We added the FCS/SSC plots in supplementary data.
Comment 12: Most of the effects on monocytes and macrophages can be explained by different cytokine levels in the supernatant, only secondary to the effect of the extract
Answer 12: We added the supplementary file(S2) containing the results of ELISA and qPCR
- Please discuss that the altered cytokine profile can explain most of the results (=indirect effect of the extract)
Answer 12’: We added the sentence for the alteration of cytokine profiles at line 310.
Comment 13: Figure 4: Polarization patterns of macrophages. The data shown is generated by FISH. Please provide information how the detection of M1 and M2 was performed since it is not stated in the material and methods section. Please also provide percentages of the populations of the total cell count. Again, the FACS patterns suggest that many cells were dead at the time of analysis
Answer 13: The information for fluorescence labeled probes was already described at 2.5 section and we revised Figure 4
- No new information on live/dead was added
Answer 13’: We added the data (FSC/SSC) at supplementary file (S1).
Comment 14: Figure 5 (the first one): LPS should induce phagocytosis. However, LPS is also present on the E. coli that are used for the experiment. Maybe the pre-treatment with LPS lead to an exhaustion of the cells? Again, a live/dead analysis would help to understand the results, because the autofluorescence of dying cells could account for the positive staining after treatment
Answer 14: Incorrect data were reflected to graph. We revised Figure 5
- Now, the phagocytosis of LPS-treated cells is even lower than in the first version although it should potently induce phagocytosis.
- No new information on live/dead was added what would be important to exclude autofluorescence
Answer 14’: We added the data (FSC/SSC) at supplementary file (S1)
All results from flow cytometry analysis were estimated with negative control to exclude autofluorescence
Comment 15: Figure 5 (the second): Please provide FCS/SSC, the panels suggest that there were not many cells left; Please explain what the right panels in a and b show
Answer 15: We revised Figure 5
- No new information on live/dead was added (now figure 6)
- No information was added what the right panels in a) and c) show
Answer 15’: We added the data (FSC/SSC) at supplementary file (S1) and explained the panels
Comment 17: Figure 7a: The control already has >70% Tregs what is unphysiologically high (normal rate is 5%). Please provide your gating strategy and back up this data with primary T cells
Answer 17: After gating with FoxP3 or NK1.1, CD304+ Treg and NKTL cells were evaluated.
We described information for the gating in legend of Figure 7
- Please provide FACS plots for the gating strategy
- It would be nice to use primary T cells to evaluate physiological effects.
Answer 17’: We added gating strategy at the legend of Figure 7.
Minor points:
Comment 24: Materials and methods line 102-103: please provide information on the clones
Answer 24: We added information of the clones
- Please provide the exact name of the clones, not only the species (or were they polyclonal?)
Answer 24’: We described the exact information at Materials
Comment 26: Materials and methods line 113: Please provide information in the asterisks used
Answer 26: Line 113 is the title for table1 but I couldn’t found the asterisks
- The asterisks are in figure 3
Answer 26’: we described the legends of Figure 3 without asterisks for accordance with other legends.
Answer 27: Figure legends: Provide information on the asterisks (**missing in figure 3, no asterisks at all in 1, 2, 4, 5, 6 and 7; does this mean all bars except the “n.s.” ones are significant?)
Answer 27: we revised the legends
- Unfortunately, I still don’t understand. Does it mean that each and every bar is significantly different from all the others? Why did you chose a different way of labeling for figure 3?
Answer 27’: Without the n.s, all bar are significant. We described the legend of Figure 3 without asterisks for accordance with other legends.
Comment 30: Interpretation of the data: If only the supernatant of macrophages is used for the treatment of monocytes and T cells: How much of the extract is still left in the supernatant? Furthermore, if the macrophages die under the treatment, less/other cytokines are produced by them and hence many effects can be explained by different cytokine levels
Answer 30:
To evaluate effects of secretory factors derived from macrophages, we treated 10ul of the supernatant to monocytes and T cells respectively and the results were compensated with results for directly treating with the same as the concentration of the extract in the treated supernatant. We added the sentence at 2.3
- Does the extract itself have any effect on the cells? If so, what effects were found? If not, the information is still important and should be discussed.
Answer 30’: In MTT assay, the left extracts (0.001μg/mL) didn’t affect for apoptosis of macrophages. We added the sentence at result section (line 258).

Reviewer 3 Report
The proposed changes have been made. I believe that the study has gained coherence and depth.
Thank you very much
Author Response
I appreciate for your detail comments. With your comments, our manuscript was improved and clarified to explain goals in this research.